# Noradrenaline and Movement Initiation Disorders in Parkinson’s Disease: A Pharmacological Functional MRI Study with Clonidine

**DOI:** 10.3390/cells11172640

**Published:** 2022-08-25

**Authors:** Marion Criaud, Chloé Laurencin, Alice Poisson, Elise Metereau, Jérôme Redouté, Stéphane Thobois, Philippe Boulinguez, Bénédicte Ballanger

**Affiliations:** 1Institute of Psychiatry Psychology & Neuroscience, Department Child & Adolescent Psychiatry, Kings College London, London SE24 9QR, UK; 2Université de Lyon, 69622 Lyon, France; 3Université Claude Bernard Lyon 1, 69100 Villeurbanne, France; 4INSERM U1028, Lyon Neuroscience Research Center (CRNL), 69000 Lyon, France; 5CNRS UMR5292, Lyon Neuroscience Research Center (CRNL), 69000 Lyon, France; 6Hôpital Neurologique Pierre Wertheimer, Service de Neurologie C, Centre Expert Parkinson, Hospices Civils de Lyon, 69677 Bron, France; 7CERMEP-Imagerie du Vivant, 69677 Lyon, France; 8CNRS UMR5229, Institute of Cognitive Science Marc Jeannerod, 69500 Bron, France

**Keywords:** Parkinson’s disease, movement initiation, akinesia, inhibitory control, noradrenaline, clonidine, α2-adrenoceptor, functional MRI

## Abstract

Slowness of movement initiation is a cardinal motor feature of Parkinson’s disease (PD) and is not fully reverted by current dopaminergic treatments. This trouble could be due to the dysfunction of executive processes and, in particular, of inhibitory control of response initiation, a function possibly associated with the noradrenergic (NA) system. The implication of NA in the network supporting proactive inhibition remains to be elucidated using pharmacological protocols. For that purpose, we administered 150 μg of clonidine to 15 healthy subjects and 12 parkinsonian patients in a double-blind, randomized, placebo-controlled design. Proactive inhibition was assessed by means of a Go/noGo task, while pre-stimulus brain activity was measured by event-related functional MRI. Acute reduction in noradrenergic transmission induced by clonidine enhanced difficulties initiating movements reflected by an increase in omission errors and modulated the activity of the anterior node of the proactive inhibitory network (dorsomedial prefrontal and anterior cingulate cortices) in PD patients. We conclude that NA contributes to movement initiation by acting on proactive inhibitory control via the α2-adrenoceptor. We suggest that targeting noradrenergic dysfunction may represent a new treatment approach in some of the movement initiation disorders seen in Parkinson’s disease.

## 1. Introduction

Although Parkinson’s disease (PD) has long been considered a motor and dopaminergic disease, numerous non-dopaminergic neurotransmitter systems are implicated in its motor features (Figure 1A) [1]. Current drugs that modulate the acetylcholinergic, glutamatergic, histaminergic, adenosinergic, GABAergic, serotonergic or noradrenergic (NA) systems might thus provide clinical benefits as add-on therapies to L-dopa by targeting symptoms that may be mediated by nondopaminergic systems [1,2,3,4]. However, the neurochemical bases of numerous motor subfunctions and non-motor functions that modulate movement control are still obscure, making it difficult to associate symptoms with potentially relevant pharmacological solutions.

Within this context, akinesia, referred to as slowness in movement initiation [5,6], has long been considered a pure motor symptom of PD, related to dysfunctions of the circuit linking the motor cortices to specific sensorimotor territories within the basal ganglia nuclei [7,8]. However, this view fails to explain experimental observations, such as the fact that lesions of the motor thalamus do not result in akinesia [9], or that internal pallidum lesions do not improve it [10]. Moreover, slowness in movement initiation is not fully reverted by dopaminergic treatments [11,12,13]. It is therefore unlikely that only a dysfunction of the motor cortico–subcortical circuit fully accounts for slowness in movement initiation.

Impairment in initiating movements in PD might also be related to executive dysfunctions [14]; in particular, to abnormal proactive inhibitory control [12,15,16]. Proactive inhibition is a pivotal mechanism which gates movement initiation in anticipation of stimulation when the context is uncertain and requires action restraint (Figure 1B) [17,18,19,20]. Proactive inhibition has been considered the default state of the executive system [19,20,21]. In healthy subjects, it takes less than 300 ms to release proactive inhibition and allow response triggering after an appropriate stimulus has been identified [21]. It takes no more time to release this default state of inhibition after a cue has removed uncertainty about upcoming stimulation, i.e., to switch from controlled to automatic behavior [22,23]. This ability would be impaired in PD patients [14], who would have difficulties releasing the default proactive inhibitory mode of control, even when the situation does not require action restraint [15]. This would substantially delay the initiation of responses, especially when the release of proactive inhibition must be internally driven (Figure 1B).

The difficulty of PD patients to release proactive inhibition is not fully compensated by dopaminergic medication [12]. Multiple clues point to the possible role of the NA system. Notably, the NA system has been involved in executive functions that likely share common mechanisms with proactive inhibition [24,25,26,27,28,29,30,31]. More specifically, clonidine, a specific α2-adrenergic receptor (AR) agonist, was found to cancel the positive action of subthalamic nucleus-deep brain stimulation (STN-DBS) on akinesia suggesting, at least in part, a noradrenergic-dependent STN-DBS efficiency in movement initiation [16,32].

As disorders of proactive inhibitory control in PD may account for various clinical symptoms ranging from akinesia (impaired ability to release proactive inhibition [15]) to impulsivity (impaired ability to sustain proactive inhibition [33,34]), increased knowledge on how noradrenaline may affect the neural network underlying this executive function may fuel the development of more optimal pharmacological treatments. Accordingly, to test the hypothesis that NA dysfunction plays a role in the pathophysiology of akinesia in PD, we manipulated NA tonus by means of clonidine, and assessed the ability of parkinsonian patients to control movement initiation with respect to healthy subjects in a double-blind placebo-controlled functional MRI study.

## 2. Materials and Methods

### 2.1. Participants

Two groups participated in the study. Fifteen healthy control subjects (aged from 41 to 70 years, six males), with no history of neurologic or psychiatric disorder, were recruited from advertisement. Twelve idiopathic parkinsonian patients (aged from 45 to 70 years, eight males), with no history of neurological disorder other than PD and current psychiatry comorbidity, were also enrolled. All participants were right-handed with normal or corrected-to-normal vision and underwent a medical screening. Only subjects with a systolic blood pressure above 100 mmHg and a diastolic blood pressure above 70 mmHg were included in the study. Exclusion criteria for the two groups were: ferromagnetic implanted materials; claustrophobia; pregnancy; a history of cholinergic or noradrenergic medications; uncontrolled hypertension; glaucoma; and scoring above 130 on the Mattis Dementia Rating Scale. All patients were tested on regular PD medication. Demographic and clinical characteristics of participants are presented in Table 1.

### 2.2. Drug Design

A double-blind, placebo-controlled, cross-over design was used with nine healthy subjects and eight patients randomized to receive a single oral dose of a lactose placebo on a first session, followed by 150 µg of clonidine on a second session, as well as six controls and five patients randomized to receive the drug first, followed by the placebo. For each subject, testing sessions were separated by at least 5 days. Each participant was tested at approximately the same time of the day (afternoon). They were instructed to abstain from caffeine, nicotine and other psycho-active substances from 24 h before the start of the session. Since peak plasma concentration for clonidine is achieved approximately 1–3 h following oral dosing [35], functional MRI sessions started 90 min after administration and lasted 1 h. Clonidine has well-established anti-hypertensive properties; accordingly, blood pressure was monitored for subject safety. Measurements were taken every 30 min, starting from drug administration until the end of the functional MRI scans. None of the participants reported any side effects of the medication.

### 2.3. Experimental Design and Apparatus

Subjects were asked to react as fast as possible to visual Go stimuli by pressing a nonmagnetic handgrip with their right hand (Figure 2; see detailed procedure of stimulus design and presentation in [15]).

In order to optimize the discriminative power of the fMRI contrast intended to reveal proactive control related activity, we used only the longest pre-stimulus delays (four to six seconds) [21]. The experiment was divided into two acquisition sessions—placebo/clonidine—with four runs each. Each run was composed of 20 Go trials, 20 NoGo trials, 20 Go_control trials and 20 catch trials (no stimulus), randomly presented, for a sum of 80 trials/condition of interest, giving a total of 320 trials for each session.

Time series from the handgrip were sampled at 1000 Hz (12 bits A/D converter). Force signals were filtered using a second-order Butterworth filter (dual pass 30 Hz lowpass cut-off frequency). Reaction times (RTs) were obtained after standard time series analyses [37]. Based on the distributions of baseline fluctuations and response peaks, movement initiation was defined as the time at which the grip force exceeded the baseline mean force signal + 35% to reach response peak force. RT was defined as the time elapsed between stimulus presentation and movement initiation. As it increases as a function of the time needed to release proactive inhibition, RT is an appropriate marker of response inhibition in this kind of task [15,19,38,39]. Inappropriate responses to NoGo signals or in absence of signal (commission errors) were analyzed to index difficulties in implementing proactive inhibition (impulsivity). Go trials without response (omission errors) were analyzed to index difficulties in releasing proactive inhibition (akinesia).

Images were acquired on a 1.5-T Siemens MRI scanner, equipped with a circular polarized head coil. For each participant, we acquired a high-resolution structural T1-weighted image (EPI sequence, resolution 1 × 1 × 1 mm) in sagittal orientation, covering the whole brain. For functional imaging, we used a T2*-weighted echoplanar sequence, covering the whole brain with 28 interleaved 3.44-mm-thick/0-mm-gap axial slices (repetition time = 2620 ms, echo time = 60 ms, flip angle = 90°, field of view = 220 cm, 64 × 64 matrix of 3.44 × 3.44 × 4.4 mm voxels).

### 2.4. Data Processing

Neuroimaging data were processed using the Statistical Parametric Mapping software (SPM8; http///www.fil.ion.ucl.ac.uk/spm/ first accessed on 1 September 2014), according to the general linear model. In order to account for magnetic saturation, the effects of the first five functional volumes of each run were removed. The other 240 images were corrected for differences in slice acquisition time. They were then realigned for the correction of head movements. Scans displaying more than 1.5% variation in global intensity, and scans showing more than 0.5 mm/time repetition in scan-to-scan motion, were considered as outliers and repaired using the ArtRepair SPM toolbox (http://spnl.stanford.edu/tools/ArtRepair/ArtRepair.html, first accessed on 1 September 2014). The DARTEL toolbox was used to perform spatial normalization on an MNI template. Data were smoothed spatially using an isotropic Gaussian filter (8mm full width at half maximum).

All events were convolved with a canonical hemodynamic response function after being time-locked to the onset of the red or green cue and modeled according to both their onset and duration. The analysis focused on the pre-stimulus period, while other events were considered as events of non-interest in the statistical analysis.

Based on the studies referenced in the introduction section, a mask encompassing all regions was found to induce potential proactive inhibition-related activity based on the aal atlas [40]. This mask includes the dorsal premotor cortex (PMd), the supplementary motor area (SMA), the dorsomedial prefrontal cortex, the inferior frontal gyrus (rIFG), the angular gyrus, the insula, the thalamus, the striatum and the STN. Data were high pass-filtered (cutoff frequency: 128 s) and summarized into one contrast per subject: the intensity of the pre-stimulus period signal was contrasted to the baseline signal intensity in each voxel.

### 2.5. Statistical Analysis

Behavioral data: Reaction time (RT) and various error rates were used as dependent variables. False alarms (responses without stimulation), abnormally short responses (RTs < 150 ms) and wrong responses (responses to NoGo stimuli) were pooled together and considered as commission errors. Missed targets (no response to Go trials) and abnormally delayed responses (RT > 1500 ms) were considered as Omission errors. The percentages of omission and commission errors were analyzed after ArcSin transforms. The experimental design was originally intended to run analyses of variance (ANOVA) with group factor and repeated measures. However, testing for data normality (Shapiro–Wilks test and Q–Q plots) and homogeneity of variances (Levene’s test) precluded applying ANOVAs. We therefore used two samples of t-tests or the non-parametric unpaired Wilcoxon test, depending on normality. RStudio 2021.09.0 was used to perform all analyses.

Event-related analysis of BOLD signal changes: In the statistical analysis, 10 event types were defined at the first level. This included two periods—pre-stimulus and post-stimulus—for five types of trial (Go_control, go, NoGo, catch_control, catch_NoGo). The events were time-locked to the onset of the cue, modeled according to their onset and their duration and convolved with a canonical hemodynamic response function. Data were high pass-filtered at 128 s and summarized into two contrasts per subject. PD patients are known to be locked into a mode of control, maintaining anticipated inhibition over willed movements even when the situation does not require proactive inhibition [15]. In other words, the green cue modality is a suitable condition for testing automatic response without inhibition in healthy subjects vs. inappropriate proactive inhibition in PD patients. To assess the interaction between drug and disease effects during the pre-stimulus period, we performed the [(green cue_(clonidine-placebo)_Patients)-(green cue_(clonidine-placebo)_Controls)] contrast. The statistical parametric group maps were generated with a random-effects model. The individual statistical maps were entered into a two-sample *t*-test: PD vs. controls. All maps were thresholded at *p* < 0.001, uncorrected for display purposes, and all results were reported after peak-level cluster-wise family wise error (FWE) correction with a significant threshold of *p* < 0.05. Finally, we also used a ROI-based analysis approach to focus on the locus coeruleus (LC), which constituted of two 10 mm spheres centered at ±4, −26 and −15 (MNI coordinates) in the pontine region of the brainstem [41,42].

## 3. Results

### 3.1. Behavioral Data

Reaction time (RT): We first tested the effect of treatment (placebo vs clonidine) for all group × task conditions. No significant effect was reported (all *p* > 0.6). We then tested the effect of task (uncertainty vs. no uncertainty) for both groups, independently of the treatment condition. The RT to Go trials were shorter in the no uncertainty than in the uncertainty condition for both groups (control group: 402 ± 74 vs. 468 ± 55 ms; t (53.3) = −3.87; *p* < 0.001) (PD group: 479 ± 86 vs. 533 ± 82 ms; W = 171; *p* < 0.05). Finally, we tested the effect of group for both conditions of task, independently of the treatment condition. The RT of PD patients were longer than the RT of control subjects, both in the no uncertainty (W = 168; *p* < 0.001) and in the uncertainty (W = 173; *p* < 0.001) conditions (Figure 3).

Commission errors: No significant effect was found.

Omissions: A significant effect of treatment (W = 1763; *p* < 0.05) revealed more omissions in the clonidine (7 ± 10%) than in the placebo condition (4 ± 6%). There was a significant effect of group in the clonidine condition (W = 237; *p* < 0.05), where PD patients produced significantly more omission errors (12 ± 15%) than controls (3 ± 5%). The effect of group just approached conventional thresholds in the placebo condition (W = 272; *p* = 0.055), whereas PD patients produced more omission errors (6 ± 10%) than controls (3 ± 7%). No other significant effect was found (Figure 4).

### 3.2. Imaging Data

By comparing PD patients to healthy controls in the no uncertainty condition, clonidine was found to increase in comparison to the placebo, with brain activation within the dorsal ACC extending to the superior medial frontal gyrus (Figure 5 and Table 2). The ROI analysis revealed a significant cluster with increased BOLD signal within the LC (including 25 voxels, *p* = 0.012 corrected at the cluster level; Z = 3.65). There was no decrease in brain activation with clonidine compared to the placebo.

## 4. Discussion

The present study provides pharmacological functional MRI evidence that noradrenaline contributes to movement initiation via the modulation of the α2-adrenoceptor and inhibitory control. The difficulty for PD patients to initiate movements with respect to controls, classically observed by RT, was enhanced under clonidine as pinpointed by an increase in omission errors. Functional MRI, assessing the specific difficulty of patients to release proactive inhibition when the situation does not require action restraint, revealed BOLD signal changes under clonidine in the LC and the anterior node of the proactive inhibitory network.

### 4.1. Noradrenergic Modulation of Movement Initiation Control

The present behavioral data confirm that PD patients not only have difficulties initiating movements when the context requires action restraint, but also have difficulties initiating movements when the context does not require action restraint. This is consistent with previous studies suggesting that PD patients are locked into a mode of control that inhibits, in advance, movement-triggering mechanisms to prevent undesired automatic responses to stimuli. In other words, PD patients would keep refraining from reacting to any upcoming stimulus, even when the situation requires automatic responding [12,15,16].

Behaviorally, clonidine did not significantly impair RT, neither in PD patients nor in healthy subjects, consistent with a previous study [44]. However, clonidine increased the rate of omissions with respect to the placebo, and this effect was more pronounced in PD patients who produced significantly more omission errors under clonidine than healthy controls. The fact that more trials were missed (with no response at all) suggests that the effect of reducing the NA tonus did not gradually impede the time needed to release inhibition (which would have been observed through the increase in RT). It is tempting to speculate that clonidine has literally prevented the release of inhibition in a substantial number of trials for which movement initiation was not possible at all, reminiscent of freezing behavior.

### 4.2. Noradrenergic Modulation of the Proactive Inhibitory Network

In the present study, BOLD changes in the LC and in the anterior node of the proactive network (mPFC/ACC) were associated with the differential effect of clonidine observed between PD patients and healthy controls in the critical experimental condition, revealing the frequently reported deficits of patients in initiating movements (Figure 5).

Previous studies already evidenced the importance of LC neuromodulation in behavioral adjustments and sensorimotor performance optimization [31,45,46,47]. Some suggested a possible role of the NA system in the ability to inhibit inappropriate movement [48,49,50,51]. However, most of this research has focused on the inhibition of movement in reaction to the presentation of a stop signal [52]. Here, our data suggested that the LC-NA system might interact with higher brain regions before any stimulus is presented to set a level of responsiveness adapted to the context. This executive function consists of modulating the tonic process, which inhibits anticipatorily, and by default, movement triggering mechanisms, which is dysfunctional in PD patients [15]. The joint modulation of activity of LC and mPFC/ACC, a central node of the proactive inhibition network [18,21], is consistent with the view that that the LC likely plays more specific roles than just autonomic arousal, and likely has a specific influence on cortical target networks supporting various cognitive and executive functions [53]. Here, our results support the hypothesis that reduced central NA activity via clonidine stimulation of the α2-ARs substantially modulates the activity of the LC and the dmFC in PD patients whose cortical NA transmission is already compromised [54,55,56]. This cortical node is the source of the ‘neural brake’ mechanism that blocks specific ongoing motor activity [57,58,59]. It is known as the “veto” area as it plays a pivotal role in intentional inhibitory control and in the initiation of voluntary action [60,61].

Although BOLD fluctuations remain difficult to interpret with regard to the potential confounds between neural excitation and inhibition [62], our observations are in line with previous results from animal studies. For instance, it was reported that stimulation of postsynaptic α2-AR produces hypoactivity in open filed tests in rats [63,64,65], while local selective blockade of those receptors in the monkey prefrontal cortex leads to increase locomotor activity and impulsivity [66]. Specifically, the BOLD signal increase observed in the dmFC in the present study is in good agreement with animal studies reporting hyperactivity of the mPFC pyramidal neurons after lesion of the LC [67], while LC stimulation has been shown to inhibit neuronal firing in the same region [68].

### 4.3. Relevance to a Noradrenergic Approach to Current Medical Therapies in PD

Functional brain measures may be more sensitive than behavioral indexes to subtle pharmacological effects [69]. Our study measured an acute effect of clonidine, while several weeks of daily treatment with clonidine were generally needed to exert its maximal beneficial effects, such as a reduction in impulsivity in patients with Tourette syndrome or with ADHD [70]. In other terms, BOLD changes observed in this study with clonidine could precede behavioral effects induced by chronic administration. Targeting non-dopaminergic medications is a major issue in PD therapy in general and is more specifically critical for akinesia, which is not fully reverted by current dopaminergic treatments [11,12,71]. The present proof-of-concept study might open the way to future clinical trials.

It has become increasingly apparent that the neuropathological changes of PD extend well beyond the nigrostriatal system, pointing especially to the early involvement of the LC in the neurodegenerative process underlying the disease [72,73]. Of interest, the NA system is thought to be involved in the pathophysiology of gait disorders in PD, such as freezing of gait which can be viewed as a failure to initiate movement [74,75,76].

On the basis of the present results, it might be relevant to explore the effect of an α2-AR antagonist in PD patients. Indeed, given that clonidine, an α2-AR agonist, negatively affects proactive inhibitory control, it is tempting to speculate that an α2-AR antagonist might conversely improve specific executive functions and behaviors, such as impulse control disorders [51,77,78,79,80,81,82]. Previously, it has been shown that α2-AR antagonists improve tremor and rigidity in the reserpinized rat [83], have a potent effect on levodopa-induced dyskinesia in a PD monkey model [84,85,86], and can extend the anti-parkinsonian effect of levodopa in MPTP-treated monkeys [87,88]. In PD patients, although conflicting results have been reported, preliminary clinical trials have suggested that this therapeutic approach reduces dyskinesia when given in combination with levodopa [89,90].

There are still, however, numerous issues to be dealt with before considering new NA drugs as potential add-on therapies to L-dopa in PD. The literature is very confusing and sometimes controversial about the potential anti-parkinsonian impact of an α2-AR antagonist. This is likely due to differences in NA subtype selectivity, differences in functional specificity with regards to neural mechanisms and to the non-selective binding of most pharmacological agents. Indeed, there are three subtypes of α2-ARs, including α2A, α2B and α2C [91,92], each with a distinct distribution and function in the brain [93]. However, the individual role of each of these subtypes is still unclear. For instance, while α2A/C antagonism (such as fipamezole and idazoxan) can potentially reduce dyskinesia in patients [89,94], working memory deficits are partly compensated by α2A-ARs agonists, such as guanfacine or clonidine [95,96,97], but overexpression of α2C-ARs can impair the ability to perform spatial and nonspatial cognitive tasks [98]. Furthermore, there are often strong confounds when using pharmaceuticals that do not target unique receptors or transporters, such as clozapine which is an α2-ARs antagonist, but which also acts as an antagonist at the dopamine D2 receptor, and modulates serotonin and acetylcholine [99]. Thus, although it has been reported to reduce LID without worsening PD [100], the anti-adrenergic properties of clozapine have never really been highlighted [3]. Accordingly, in vivo imaging of all receptors could further elucidate the potential important roles of α2 ARs in PD patients, not only in movement control but also in the context of cognitive decline and non-motor symptoms. This is a current technical challenge for molecular imaging [101].

## 5. Conclusions

Although no convincing solution has been provided to date [2,4], the present results might have important implications in setting the ground for new add-on NA treatment approaches. Yet, further pharmacological investigations are warranted to support this hypothesis. Recent technological and methodological developments in molecular imaging might offer new opportunities to better understand the role of NA and adrenoceptors in neurodegeneration, a central issue in PD pathophysiology [101,102,103,104,105,106,107]. A condition for success is certainly to combine molecular imaging with clinical and behavioral analyses, yielding refined functional segregation [5,108]. Of particular interest is the new possibility to reliably investigate in α2 density in vivo in large human samples [109]. Understanding the NA pathogenic mechanisms that might contribute to disease progression and associated complications is indeed a prerequisite for developing novel neuroprotective or efficient disease-modifying therapies with individually tailored care.

## Figures and Tables

**Figure 1 cells-11-02640-f001:**
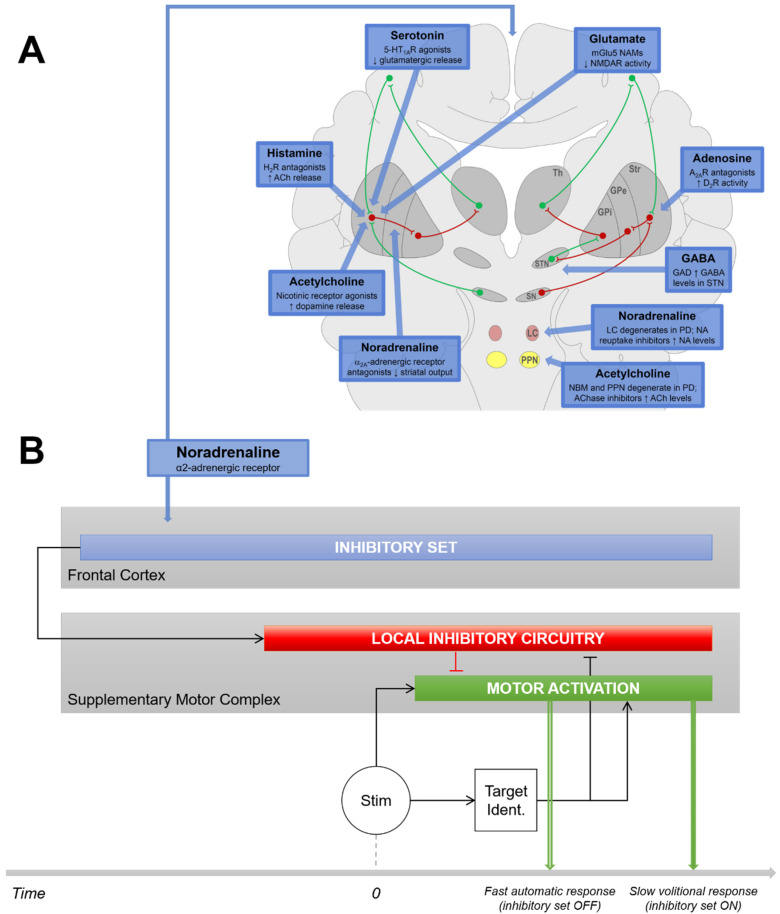
Issues and Hypotheses. (**A**) Nondopaminergic neurotransmitter systems involved in the motor features of PD. Taken from [1] and reproduced with permission. (**B**) We hypothesize that the impairment in initiating movements in PD patients might be related to the NA system. Indeed, the NA system likely plays a substantial role in proactive response inhibition: a cortico–ganglio–thalamo–cortical function is intended to inhibit movement triggering mechanisms by anticipation to prevent erroneous responses when the context is uncertain. When proactive inhibition is ON, motor responses are delayed with respect to a condition that does not require inhibition (fast automatic responses) because it takes additional time to release inhibition after the stimulus has been identified. PD patients are known to have enhanced difficulties to trigger automatic responses when the context does not require action restraint. This might be due to the fact that PD patients are often locked into a mode of control, maintaining inappropriate proactive inhibition over willed movements (i.e., troubles to switch from controlled to automatic behavior). If this disorder is associated with the NA system, manipulating noradrenergic tonus by means of clonidine, an α2-AR agonist, should induce brain activation differences in the proactive inhibition network associated with the lengthening of reaction time in PD patients with respect to healthy controls. GPe, globus pallidus externa; GPi, globus pallidus interna; LC, locus coeruleus; SN, substantia nigra; Str, striatum; STN, subthalamic nucleus; Th, thalamus.

**Figure 2 cells-11-02640-f002:**
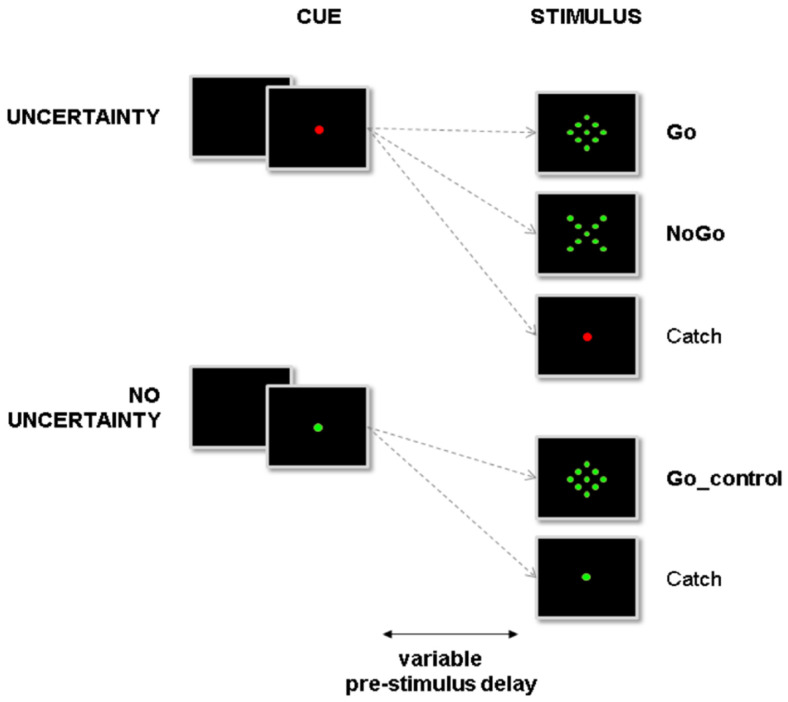
Overview of the experimental setup. Subjects were instructed to react to the presentation of a go signal (diamond) by pressing a button. At the beginning of a trial, the central fixation point (cue) could turn either red or green, indicating, respectively, that NoGo stimuli (X) could or could not be presented. In the former condition (uncertainty condition), proactive inhibition was required during the pre-stimulus period to avoid erroneous automatic responses to NoGo stimuli. In the latter condition, proactive inhibition was not required during the pre-stimulus period. Subjects could react automatically to any upcoming target (no uncertainty condition).

**Figure 3 cells-11-02640-f003:**
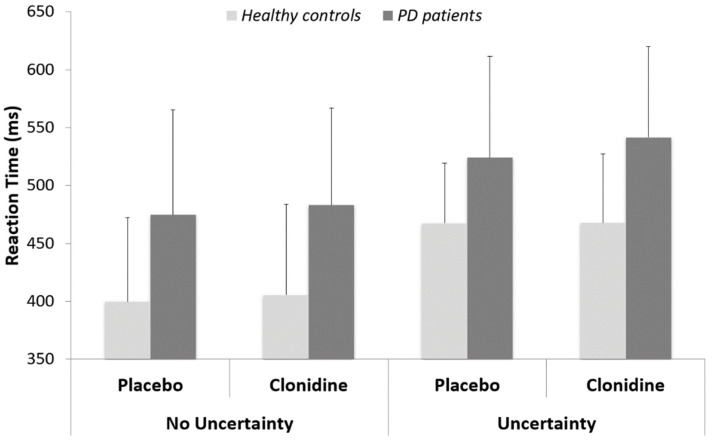
Reaction times (means and standard deviations) for PD patients versus healthy matched controls.

**Figure 4 cells-11-02640-f004:**
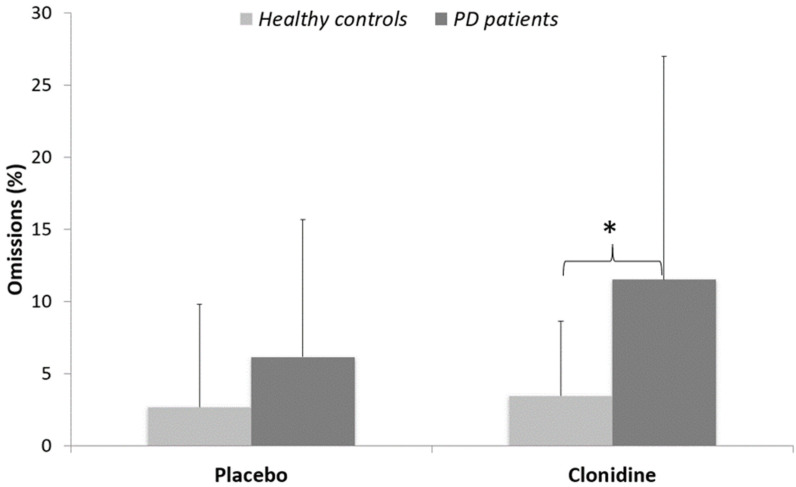
Percentages of omissions (means and standard deviations) for PD patients versus healthy matched controls. * *p* < 0.05.

**Figure 5 cells-11-02640-f005:**
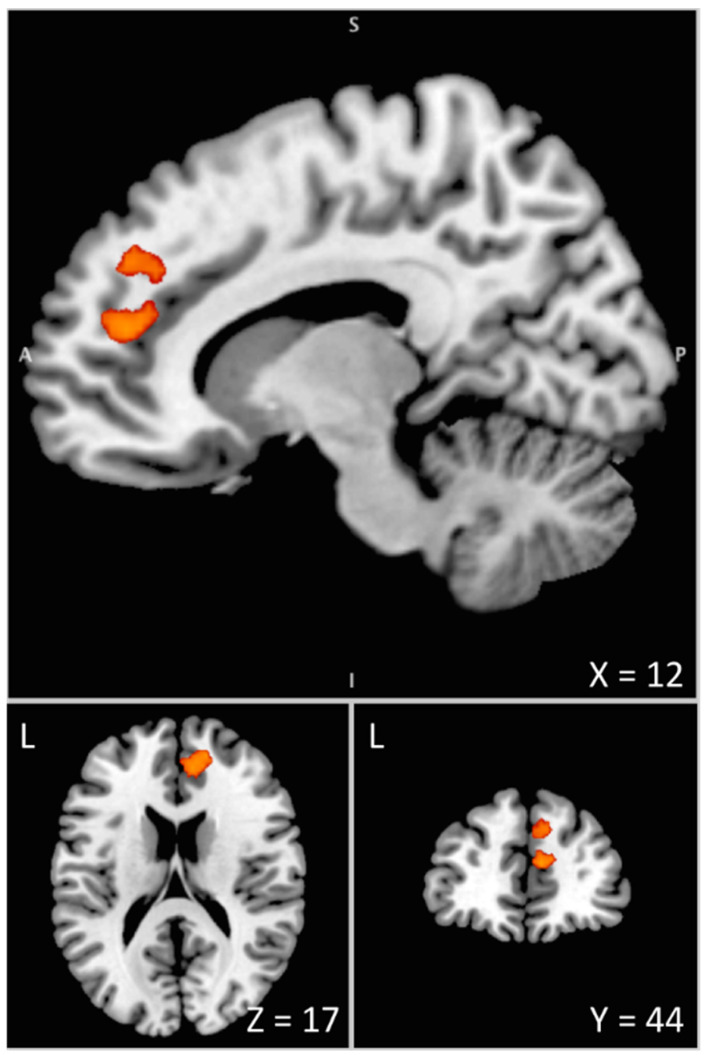
Interaction between drug and disease effects during the pre-stimulus period assessed by means of the [(green cue_(clonidine-placebo)_Patients)-(green cue_(clonidine-placebo)_Controls)] contrast. The differential increase in BOLD signal under clonidine between PD patients and healthy controls is overlaid on the Colin 27 brain template in the MNI space visualized with the Mango software [43]. This overactivation of the anterior node of the proactive inhibition network (mPFC/ACC) is associated with the enhanced difficulty of PD patients to initiate movements when action restraint is not required (control condition with no uncertainty) under the effect of clonidine. L = Left. A = Anterior. S = Superior. X, Y, Z are coordinates in mm in the MNI space.

**Table 1 cells-11-02640-t001:** Demographic and clinical characteristics of the participants.

	Healthy Controls (*n* = 15)	PD Patients (*n* = 12)	Group Difference
Age in years (SD)	52.5 (11.2)	56.2 (8.9)	ns
Male/female	6M/9F	8M/4F	-
UPDRS-III (ON medication)		12.7 (4.8)	-
Disease duration in years (SD)		6.1 (2.3)	-
LED (mg/day)		948 (320)	-
Mattis	141 (3)	137 (3)	*p* < 0.006
Beck Depression Inventory	3.9 (3.3)	11.1 (6.2)	*p* < 0.0005

Data are presented as mean (SD, standard deviation); ns, not significant; UPDRS = Unified Parkinson’s Disease Rating Scale; LED = Levodopa Equivalent Dose (calculated according to [36]). Two sample *t*-tests were used to compare demographic and clinical variables between groups.

**Table 2 cells-11-02640-t002:** Location of increased BOLD signal in patients with PD versus healthy controls during the control condition (with no uncertainty) under clonidine versus placebo.

			MNI Coordinates				P Corr	Cluster
Areas	BA	Side	x	y	z	Z-Score	Cluster	Size
Anterior Cingulate Cortex	32	R	8	44	16	4.21	0.037	311
*Superior Frontal Gyrus*	*32*	*R*	*16*	*50*	*18*	*4.03*		
*Superior Medial Frontal Gyrus*	*9/32*	*R*	*10*	*44*	*34*	*3.72*		

R = right; BA = Brodmann’s area.

## Data Availability

The authors will be happy to support request for a formal data sharing agreement.

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
