# Peer review of "Noradrenaline and Movement Initiation Disorders in Parkinson’s Disease: A Pharmacological Functional MRI Study with Clonidine"

_cells, 2022, doi:10.3390/cells11172640_

Round 1
Reviewer 1 Report
In this manuscript the authors present an interesting hypothesis on akinesia in Parkinson's disease. Although the topic is not new, the approach using functional MRI seems interesting.
My main concerns are
1.the sample size and the tool used for recruitment of controls.
2.the reading of the manuscript is complex and needs clarification on the hypothesis to be tested. I suggest adding a figure for better understanding of the noradrenergic pathways involved.
3. Although the results support the main objective, some previous evidence needs to be discussed. For example, the role of atypical neuroleptics such as clozapine, with an alpha2-adrenergic antagonism, which indeed showed controversial results, on the one hand clozapine may improve tremor, as proposed by the authors, but on the other hand a negative impact on movement in PD may occur.
Author Response
Response to Reviewer 1 Comments
In this manuscript the authors present an interesting hypothesis on akinesia in Parkinson's disease. Although the topic is not new, the approach using functional MRI seems interesting.
My main concerns are
Point 1: the sample size and the tool used for recruitment of controls.
Response 1: The sample size was calculated on the basis of the expected differences informed by former behavioural results using a similar design in PD patients and healthy controls testing for the effect of dopaminergic treatment (Favre et al 2013). We used this dataset to characterize the expected difference in RT, the main dependent variable pinpointing proactive inhibitory control, between both groups in the main condition of interest (no uncertainty) under placebo. Based on other recent results (Albares et al. 2015), we expected Clonidine to increase this difference.
Given a desired power rate of 80%, and a predetermined Type 1 error rate α of 5%, the number of patients required to evidence a RT difference between PD patients and controls of 70 ms (with expected mean under H0=313 and expected SD=70), the minimum number of subjects to be included was 26 (13 in each group). We initially planned to include 15 patients within each group, but due to limitations of the imaging platform, we could test only 12 PD patients before the end of the time allowed to perform the experiment constrained by the pharmacological procedure. We thus checked a posteriori for the validity of this new unequal sample size using the actual data for calculation, which returned a minimum group size of 15 controls for 12 patients (Given a desired power rate of 80%, a predetermined Type 1 error rate α of 5%, a RT difference between PD patients and controls of 75 ms, a mean under H0=400 and a SD=81).
Control subjects were recruited from advertisement and screened by a neurologist (ST) to check for inclusion and exclusion criteria. Control subjects have not been paired individually to PD patients, but we controlled for age by testing differences between groups using a two-sample t-test. Missing information has been added in the text and in Table 1 legend.
Point 2: the reading of the manuscript is complex and needs clarification on the hypothesis to be tested. I suggest adding a figure for better understanding of the noradrenergic pathways involved.
Response 2: We agree with Reviewer 1. First, we have added a paragraph (lines 38-46) in the introduction section in order to better include our working hypothesis within the theoretical framework of nondopaminergic therapeutic solutions. Second, we have also added a figure (Figure 1) as requested to clarify our hypothesis centered on the frontal NA pathway.
Point 3: Although the results support the main objective, some previous evidence needs to be discussed. For example, the role of atypical neuroleptics such as clozapine, with an alpha2-adrenergic antagonism, which indeed showed controversial results, on the one hand clozapine may improve tremor, as proposed by the authors, but on the other hand a negative impact on movement in PD may occur.
Response 3: We fully agree with the referee that the literature is very confusing and sometimes controversial about the anti-parkinsonian impact of an α2-AR antagonist. This is likely due to differences in NA subtype selectivity, differences in functional specificity with regard to neural mechanisms, and to the non-selective binding of most pharmacological agents.
This is the case for clozapine. It has been reported that clozapine significantly reduces LID without worsening PD (Durif et al., 2004). But clozapine has mixed binding properties involving among others serotonin and acetylcholine. In fact, its anti-adrenergic properties have never really been highlighted (Fox et al., 2013). In addition, the clinical use of this therapeutics for LID and/or PD tremor is rare in practice because of the mandatory blood monitoring needed to prevent agranulocytosis (Fox et al., 2013).
These points have been discussed in the appropriate section (lines 396-416). We thank Reviewer 1 for raising this important issue.
Reviewer 2 Report
In the present study Criaud, Laurencin, Poisson and colleagues investigate proactive inhibition using functional MRI. Administration of clonidine in PD subjects induces reduction of noradrenergic transmission leading to movement initiation issues, possibly through inhibition of a2-adrenoceptor.
The manuscript is easily readable and well written, but some changes need to be addressed:
Major comments:
· Results sections: the authors describe the results in a narrative way, but those should be also plotted for visualization.
Please provide also graphs.
· Authors in describing the results do not explain if they are shown with ± SEM or SD.
· The authors should also include in the text the degree of freedom and all the necessary descriptive statistic information.
· Figure 2: it is not clear if the represented MRI is from a healthy or PD subject. Is this an integrated control versus PD patient MRI image?Please provide a representative image of both PD and healthy patients. Also provide a raw figure without the overlay mask next to the one with the overlay mask.
· The authors used only right-handed patients. Is there a specific reason for this or is it only due to the availability of right-handed subjects? Do the authors believe the response could be different in left handed patients? Could they speculate right/left handed differences this in their text?
· Why if clonidine needs weeks to start being fully effective the authors investigated its effect minutes after its administration? Wouldn’t be the cognitive performance more accurate after a few weeks?
Minor comments:
· Please give more detail on the parkinsonian patients. Are those all sporadic PD patients? Even the ones aged 45? Has a genetic screening been done for PD-associated genes/risk variants?
· Authors should stick to one style of presenting significance e.g., 0.05 or .05
· A conclusion/summary section is missing.
· Line 140: reaction time (RT) needs to be introduced there and not in line 145.
· 261-267: references are missing. One is not enough
Round 2
Reviewer 1 Report
The present manuscript have been improved and a I suggest it publication in the present form